# Challenges and Adaptations for Resilient Rice Production under Changing Environments in Bangladesh

Md Roushon Jamal [1,2], Paul Kristiansen [1,*], Md Jahangir Kabir [3] and Lisa Lobry de Bruyn [1]

1   School of Environmental and Rural Science, University of New England, Armidale, NSW 2350, Australia;
    roushonjamal@yahoo.com (M.R.J.); llobryde@une.edu.au (L.L.d.B.)
2   Department of Agricultural Extension, Ministry of Agriculture, Dhaka 2202, Bangladesh
3   UNE Business School, University of New England, Armidale, NSW 2350, Australia;
    jahangir.kabir@une.edu.au
*   Correspondence: paul.kristiansen@une.edu.au

**Abstract:** Rice-based food production is crucial for food security, socio-political stability, and economic development in Bangladesh. However, climate and environmental changes pose serious challenges to sustainable rice production in the country. This review paper critically reviews the status, challenges, and adaptation opportunities of Bangladeshi rice systems in changing environmental, demographic, and socio-economic settings. A mixed-methods approach (quantitative summary of secondary data on rice production, environmental changes, impact on rice productivity; and qualitative thematic synthesis) was used to review adaptation for resilient rice production. Significant agronomic and environmental challenges to rice-based cropping in Bangladesh are posed by rising temperatures, anomalous rainfall patterns, extreme weather, and increasing salinisation. Rice production, availability, and access have been further compromised by decreasing arable areas, labour shortages, crop diversification, and low profitability. Farmers are adapting through autonomous and centrally planned strategies such as efficient irrigation and input use, stress-tolerant cultivars, mechanisation, and income diversification. However, profitable and sustainable adaptation requires broader facilitation by the government (e.g., infrastructure, financial incentives) and agribusiness (e.g., extension services, contracting). This review paper recommends research and development support for efficient irrigation management and stress-tolerant cultivars, enabling policy initiatives, and equitable value chain participation. The insights of the review can be applied to policymakers to target policy design and decision-making for a sustainable rice system in years to come.

**Keywords:** food security; rice-based farming systems; autonomous adaptation; procurement policy; climate-smart technologies

## 1. Introduction

A sustainable rice production system is a global concern under changing environment, market, and socio-economic dynamics, as rice is the staple food of more than 50% of the global population [1,2]. This dominant cereal crop is also vital for global food security, as it is consumed by the majority of the global poor [3]. Rice is the most widely consumed food crop, with an annual production of 509 million tons (Mt) from 16 million hectares (ha) of rice area in over 100 countries in 2020 [4]. As a strategic food commodity due to its importance in food security, income, and employment, governments in many countries intervene in rice production, stock, and supply with input subsidies, incentives, imports, minimum support price, public procurement, and trade policies [5]. Irrigated rice supplies 75–80% of global rice production [6]. However, the scarcity of freshwater in rice-growing countries could adversely affect targets to achieve sustainable development goals [7]. Global rice demand is projected to increase due to the rising global population and increasing per capita consumption in Sub-Saharan Africa [8], the Middle East, and South America [3].

Climate change is leading to global warming [9], declining water resources [10], salinisation [11], and land use change [12] globally and in Bangladesh. Along with economic and market changes [13] and population and demographic pressures [14], the effects of climate change are creating significant challenges to achieving sustainable development goals (SDG 1 and SDG 2) and food security. The Food and Agriculture Organization reports that over 820 million people in the world suffer from hunger, while about 2 billion people experience moderate or severe food insecurity [15]. Recently, the FAO estimated that food production needs to be increased by 60 per cent to feed the global population of 9 billion people in 2050 [4]. Resilient rice systems are essential for developing and developed countries against the backdrop of global food insecurity, environmental changes, and population pressure [16].

In Bangladesh, rice is often considered to be a political and strategic commodity because of its socio-economic, cultural, and political importance [17]. Food security largely depends on the availability of and access to rice, as rice forms 70% of people's daily calorific requirement and 56% of their protein intake [18]. Rice also occupies 78% of the total cropped area in Bangladesh [18], with an annual production of 36.3 Mt and an average annual import of 0.5 Mt [19,20]. Rice is also an important economic crop, contributing 46% of the crop sector's gross domestic product (GDP) and 5% of the overall GDP in Bangladesh [21]. Therefore, the overall food security of Bangladesh largely depends on the sustainable rice production system.

However, environmental risk and vulnerability dominantly threaten rice production in Bangladesh and other rice producing countries. Bangladesh has been ranked seventh in the 'Global Climate Risk Index-2018' because of its geographic and socioeconomic vulnerability [22]. Bangladesh is placed in 76th position in the 'Global Hunger Index 2021' with 19.1 scores [23]. The capacity to adapt to the changing environment is closely aligned with many of the sustainable development goals (SDGs) [24], especially those relating to ending hunger (SDG 2), economic growth (SDG 8), life on land (SDG 15), and climate action (SDG 13). It is evident from historical data analysis, empirical research, literature reviews, and modelling that climate variability and extreme weather events, as well as socio-economic changes, can significantly impact the rice system and food security in Bangladesh [25].

In recent decades, Bangladesh has nearly achieved rice self-sufficiency through domestic rice production [26]. However, despite increasing production, there are still major, diverse challenges ahead, posing immediate and long-term risks to rice availability [27]. About 40 million people in Bangladesh remain severely food insecure and another 11 million suffer from acute hunger [28]. Achieving the production target of 47 Mt in 2050 from declining rice areas might be a big challenge at the farm level (Figure 1).

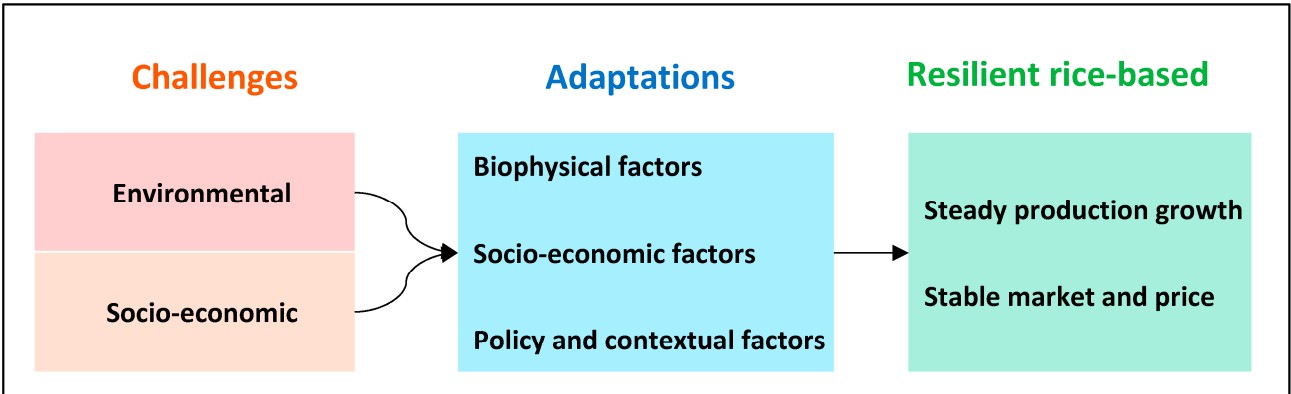

**Figure 1.** Conceptual framework of challenges and adaptation measures affecting resilient rice-based food system in Bangladesh.

Rising temperatures [29], anomalous rainfall patterns [30], more frequent extreme weather events [31], increasing salinisation [32], and the scarcity of fresh water for irrigation [12] have threatened the stability of the rice production system in Bangladesh. The predicted climate change scenario (increased temperature, increased monsoon rainfall) is likely to affect all rice-growing zones of the country, generating food security challenges and social unrest [17]. Rice farming systems have gone through many changes over the last couple of decades in response to these challenges [33]. Demographic changes such as urban migration for work and business opportunities, reduced farm labour availability, market instability, and affluence have also affected the resilience of the rice system in Bangladesh [17]. Weak public procurement policy and insufficient public storage capacity also raise concerns about not being efficient enough to buffer the supply and price shock [34]. Recently, the importation of rice into Bangladesh has been on the rise to stabilise domestic supply and markets [35].

If domestic rice production is constrained and the reliance on imports increases, this key food commodity's availability and affordability will become an issue for the rural poor of Bangladesh [17]. There is minimal scope to increase the rice production area in Bangladesh, but the overall rice production needs to be increased to meet rising domestic demand. For the rice requirement of the predicted 215 M population in 2050, Bangladesh will require more than 47 Mt of rice from a decreasing rice-production land area. The government has declared 'Rice Vision 2050' for a resilient rice system in Bangladesh [36]. Therefore, there is ongoing pressure to increase the rice yield by land remediation, increasing cropping intensity, employing modern agronomic practices, and growing high-yielding varieties [37]. However, although many adaptations have been reported in the literature, it is unclear what the preferred adaptive pathways are for farmers and how this can be supported.

Climate-smart agriculture' has emerged as an integrative framework for adapting agricultural systems to the challenges of global warming and changes in rainfall patterns [38], though there are questions about the suitability of this framework for addressing local agronomic and food security needs [29]. Therefore, understanding the challenges and adaptation responses for resilient rice production in a changing environment is imperative to sustaining future food security in Bangladesh [17].

Although several studies have been undertaken on climate change impacts on food security, an analysis of the research on the resilience of the rice production system under changing environmental and socio-economic conditions is lacking, particularly in Bangladesh. The objectives of this paper were to (1) provide a summary of current data regarding rice production challenges and impacts in Bangladesh, (2) conduct a qualitative thematic analysis of literature regarding the type of adaptation strategies, and (3) identify research, development, and extension needs. The paper begins with an overview of the current situation for rice production, consumption, and availability in Bangladesh, followed by an evaluation of historical and projected biophysical changes related to climate change salinisation, socio-economic challenges, and the known and predicted impacts on the rice market's stability. Secondly, drawing on the conceptual framework in Figure 1 (e.g., climatic and socioecological challenges, adaptations), the literature on adaptation strategies, management technologies, institutional capacity (e.g., non-government organisations [NGO] and government agencies), and policy support are reviewed. Finally, the paper summarises these key issues about the research gaps, highlighting research, development, and extension needs.

## 2. Methodological Approach

The methodology used in this review paper consisted of a mixed-methods approach with two related strategies: (1) a quantitative summary of secondary data related to rice production, environmental changes, and their impact on rice productivity and (2) a qualitative thematic synthesis [39,40]. This approach was chosen, rather than a formal systematic review, as the aim was to conduct a thematic analysis of readily available research publi-

cations, not an exhaustive review of work from a structured search. Standard systematic review methods may be less well suited to the analysis of complex systems, where the source material may include grey literature, screening may be (inadvertently) subjective, research methodologies are less uniform and often multi-disciplinary, the explanatory and response variables are less distinct, and the findings are poorly suited to quantitative evaluation [41,42].

The thematic literature synthesis was conducted to provide an in-depth narrative synthesis of available published findings on the research topic, structured according to themes that were pre-determined and revised based on themes emerging from the analysis. The literature review commenced by conducting searches on databases including Web of Science, Scopus, and Google Scholar. The search string, "resilience OR sustainability OR adaptation OR rice-based farming system OR Bangladesh" AND "climatic change OR environmental change OR salinity OR drought OR flood" AND "rice price OR rice markets" AND "challenges OR impact OR risk", was developed in alignment with the study's objectives. The search encompassed the title, abstract, and keywords of the publications, and duplicate entries were carefully eliminated from the search results. These search lists were filtered based on a careful review of the title and abstract of each publication to align with the themes of rice production, climate change and salinity impacts, adaptation, and capacity building for agricultural resilience in Bangladesh. Further literature was included in the review based on the need to contextualise the key issues within each theme (e.g., impacts in other deltaic regions, agricultural adoption dynamics, global and socio-economic trends). This literature has information about context and processes that is critical to understanding complex social and land use issues and is poorly addressed using only structured "systematic" search and filter methods [41]. Before finalising the selection of publications, including journal articles, conference papers, and government publications, a thorough reading of the full text of each publication was conducted. To keep the number of publications within a reasonable limit (150–200), the most relevant and recent ones were chosen. Quantitative publications on rice production, price, availability, climatic impacts, and salinity were collected from national (e.g., BBS, BDP, IRRI, DAE) and international (e.g., WB, FAO, IFPRI) organisations. To contextualise themes with other rice-producing countries and provide a wider perspective beyond Bangladesh, recent publications from other rice-growing regions were used.

## 3. An Overview of the Current Situation for the Rice System in Bangladesh

The dominance of rice in people's diets in Bangladesh has made the rice production system a crucial issue. Annual per capita rice consumption (Table 1) in the country (169 kg per capita) is higher than the largest rice producer, China, and leading rice exporters, Thailand, Vietnam, and India [8]. This heavy dietary reliance has made rice a strategic and politically important food crop for Bangladesh.

**Table 1.** Annual rice consumption per capita for selected countries in Asia [8].

| Country | Rice Consumption (kg per Capita per Year) |
| --- | --- |
| Bangladesh | 169 |
| Indonesia | 163 |
| Thailand | 142 |
| Philippines | 122 |
| China | 77 |
| India | 74 |
| World average | 57 |

Nevertheless, the national average yield of rice in Bangladesh is still low (3.12 t/ha) compared to other leading rice-producing countries (Table 2). Globally, Bangladesh ranks third in total rice production and consumption [4]. Despite using a considerable amount of fertiliser (282 kg/ha), the rice yield is lower than in Indonesia, where fertiliser use is 181 kg/ha [43] (Table 2). Factors contributing to lower rice yields include lack of farmers'

knowledge of modern rice management practices, slow adoption of BRRI-recommended rice-production technologies, biophysical issues such as climatic stresses (e.g., drought), and variable soil types, as well as contextual limitations, such as inadequate extension services [44].

**Table 2.** Rice production statistics in major rice-producing countries [4]. Fertiliser use refers to combined fertilisers containing nitrogen, phosphorus, potassium, and sulphur.

| Countries | Rice Area (M ha) | Irrigated Area (% of Rice Area) | Use of Modern Varieties (%) | Rice Yield (t/ha) | Fertiliser Use (kg/ha) |
|---|---|---|---|---|---|
| China | 30.1 | 100 | 100 | 4.39 | 488 |
| Vietnam | 7.5 | 95 | 96 | 3.56 | 402 |
| Indonesia | 13.2 | 94 | 92 | 3.36 | 181 |
| Bangladesh | 11.6 | 73 | 79 | 3.12 | 282 |
| Philippines | 4.3 | 69 | 97 | 2.43 | 140 |
| India | 36.9 | 60 | 85 | 2.26 | 168 |

During the last 45 years (1973–2017), the average annual growth in rice production was 2.4% (Figure 2) [19]. Despite some notable increases in annual rice production growth, the overall trend has flattened over the period shown ($p = 0.183$). Rice production growth has declined by almost 1% annually ($p = 0.003$) [19,43] in the last ten years, which is particularly concerning (2008–2017). This declining trend has led to calls for greater adoption of sustainable measures to maintain or increase rice yields nationally [17].

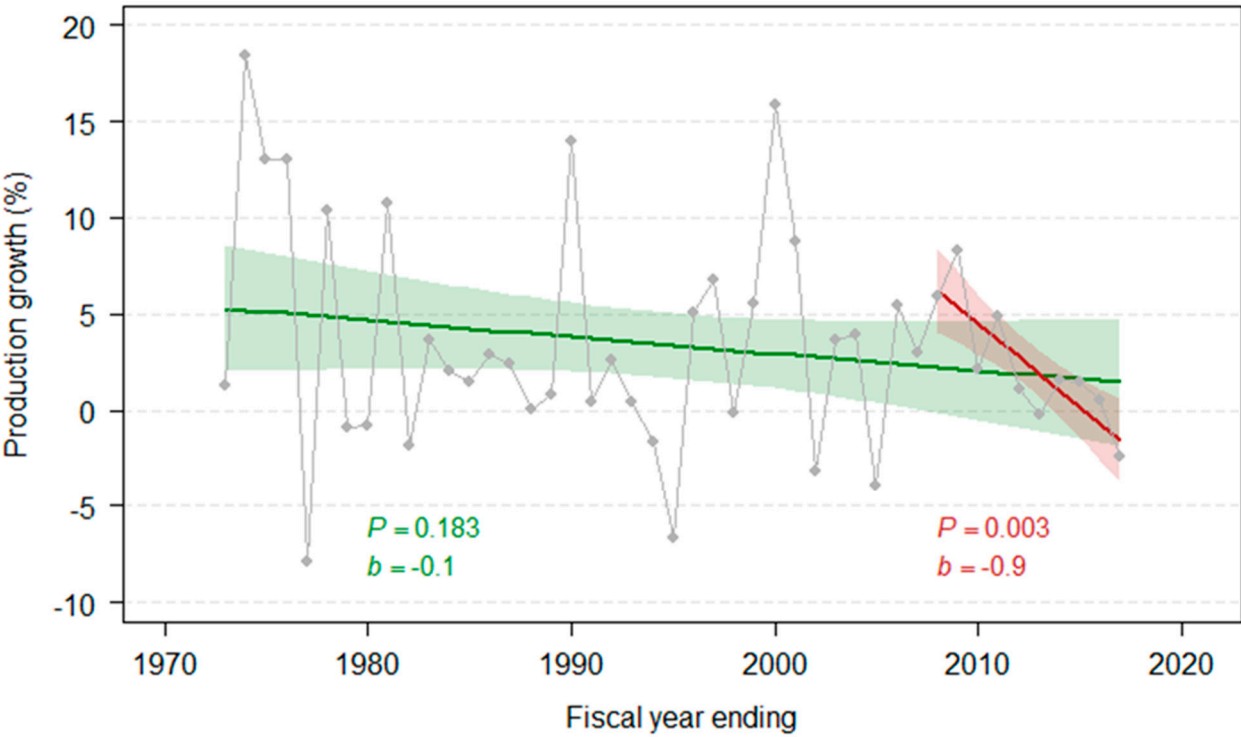

**Figure 2.** Annual growth in rice production in Bangladesh from 1972–1973 to 2016–2017 [19]. Linear regression for the full period (green) and for the last 10 years of the data set (red), shown with *p* value, slope coefficient (b), and 95% confidence bands.

The scope of increasing rice area is limited in Bangladesh. The net rice-growing area of Bangladesh is 6.46 M ha, representing 78% of the total cropland. The three rice growing seasons in Bangladesh are Aus (early wet season), Aman (wet season), and Boro (dry season). Despite a sharp increase in irrigated Boro rice cultivation from 0.86 M ha in 1971–1972 to 4.5 M ha in 2016–2017, the area under rain-fed Aus rice (3.0 M ha in 1971–1972

and 0.94 ha in 2016–2017) and Aman rice (6.01 M ha in 1981–81 and 5.58 M ha in 2016–2017) has been decreasing [19]. The net rice area is projected to decrease further due to climatic stresses, hydrological challenges, population growth, urbanisation, and crop diversification for economic reasons. More than 50% of the rice area is vulnerable to environmental stresses such as drought, early flash floods, scarce freshwater, and anomalous rainfall [45].

Nevertheless, the rice yield varies in different ecosystems based on climatic variations, soil fertility, land type, and the adoption status of modern rice technologies. Table 3 summarises the yield of Aus, Aman, and Boro rice in major rice ecosystems. Despite the good availability of fresh irrigation water, the south-central coastal zone shows the lowest yield (3.41 t/ha) of Boro rice. Rice yields in the salinity-prone south-west coastal zone is still better compared to the national average and other ecosystems. Yet, the yield of rain-fed Aus and Aman rice is low in all rice ecosystems. Though experiencing hydrological stresses (e.g., the drawdown of groundwater, drying up the surface water source), the north-west region is still showing better Boro yield.

**Table 3.** The yield of rice (tonnes/ha) in different rice ecosystems in Bangladesh [19].

| Rice Ecosystems | Boro (Irrigated) | Aman Rice (Rain-Fed) | Aus Rice (Rain-Fed) |
|---|---|---|---|
| South-west coastal zone | 3.90 | 2.58 | 2.25 |
| South-central coastal zone | 3.41 | 1.84 | 1.79 |
| South-east coastal zone | 3.70 | 2.48 | 2.01 |
| Low-lying Haor basins | 3.72 | 2.27 | 2.51 |
| Drought-prone north-west area | 4.08 | 2.55 | 2.48 |
| Central Upland area | 4.41 | 2.22 | 2.05 |
| Hill tract | 3.50 | 2.75 | 1.73 |
| Deep-water Aman ecosystem (broadcast) | - | 1.17 | - |
| Highland Aus (broadcast) | - | - | 1.52 |
| National Average | 3.96 | 2.38 | 2.22 |

The yield gap of popular rice varieties is another concern for the rice system in Bangladesh. With the limited scope for increasing cropping intensity, which is already high (at least two crops per year) in Bangladesh, future self-sufficiency in rice will largely depend on the degree of yield gap closure on existing land areas [46]. There is a considerable yield gap for the most popular varieties developed by the Bangladesh Rice Research Institute (BRRI) and Bangladesh Institute of Nuclear Agriculture (BINA). The average yield potential of most of the BRRI- and BINA-released popular high-yielding Boro rice varieties ranges from 4.5 t/ha to 7.5 t/ha [47]; however, the national yield of Boro rice is still 3.96 t/ha [19], suggesting that farmer adoption of high-yielding varieties and management practices is constrained [17]. Options for bridging the yield gap include continued work on improving rice varieties (e.g., abiotic and biotic stress tolerance, higher yield potential, grain quality, locally suited cultivars), adoption of climate-smart technologies, and curtailing adoption lag through training [46,48].

## 4. Availability of Rice in the Context of Changing Environment

The availability of food denotes the amount of food present in a country/area through domestic production, imports, food stocks, and food aid [49]. Availability is the most important pillar of food security. The present global rice area is 162.9 M ha and rice production is 594.7 Mt, with a global average yield of 3.1 t/ha [8]. Globally, per capita rice availability is 53 kg per year. Asia accounted for about 86% of the global rice area and contributed about 90% to global rice production. Global rice production needs to be increased by 25–30% to meet the rice demand in 2050. Rice availability is influenced by production, human consumption, non-food use, socio-economic drivers, global trade policy, import, and food aid [3].

In Bangladesh, the availability of rice is strategically important due to the people's heavy reliance on it for calories and protein. Rice production is frequently impacted by

climatic change, salinisation, and extreme events. The total rice area in Bangladesh is expected to be reduced by 15–20% because of the intensification of non-rice crops, non-farm activities, and population pressure. Doubling productivity concerning rice is a target for meeting SDG 2.3.1. Therefore, to meet the rice demand in 2050, the present yield (3.2 t/ha) must be raised to 5.3 t/ha [50].

Table 4 summarises the rice availability in Bangladesh over the last 20 years. Per capita, rice availability increased from 170 kg in 2000–2001 to 218 kg in 2020–2021. However, the stability of the domestic rice market is volatile with the fluctuating retail price.

**Table 4.** Rice availability (production, import, and aid) and consumption (2000–2020). Source: [18,35,51].

| Sources | 2000–2001 | 2010–2011 | 2020–2021 |
|---|---|---|---|
| Domestic production (Mt) | 25.1 | 33.5 | 39.6 |
| Import (Mt) | 0.53 | 1.56 | 2.65 |
| Food aid (Mt) | 0.03 | 0.0 | 0.0 |
| Total availability (Mt) | 25.6 | 35.1 | 42.3 |
| Non-food consumption | 3.84 | 5.26 | 6.33 |
| Net availability of rice (Mt) | 21.8 | 29.8 | 35.9 |
| Per capita availability (kg/person/year) | 170 | 202 | 218 |

## 5. Environmental Challenges for the Rice Production System

### 5.1. Climate Change

The rising global atmospheric temperature and associated changes in rainfall patterns, hydrology, and extreme climatic events are now a global concern [52]. As a vulnerable country, Bangladesh is likely to face more severe consequences of climate change. Table 5 shows the changes in historical and predicted climate in Bangladesh. During the 20th century, increased mean temperature and annual rainfall have been observed. The 21st century is likely to experience hotter summers, warmer winters, heavier rainfall events, more flooding during the wet season, drought, and extreme climatic events.

**Table 5.** Historical and projected climate change in Bangladesh.

| Historical and Projected Climate Change | Source |
|---|---|
| A 0.4–0.8 °C increase in global temperature from 1901 to 2001. An increase in annual mean temperatures of 1.0 °C by 2030, 1.4 °C by 2050, and 2.4 °C by 2100. The country will experience a 5–6% increase in rainfall by 2030. | IPCC [53] |
| From 1967 to 2007 (40 years), Bangladesh experienced a rising trend in the mean temperature of 0.40 to 0.65 °C. | Nishat and Mukherjee [54] |
| An 0.4 °C temperature increase in 50 years (1966–2015) and an increase in rainfall of 6.7 mm/year except for winter (November-February). The extreme rainfall events (3 days or more) are likely to increase in southern and eastern Bangladesh. | Mullick, Nur [52] |
| The mean temperature has increased by 0.097 °C per decade in the last 50 years—a significant change in annual mean rainfall and pre-monsoon rainfall. | Shahid [55] |
| Winter temperature is projected to rise by 1.49 °C by 2030 and 4.12 °C by 2075 relative to the base year 1990. Summer warming would likely be 0.87 °C by 2030 and 3.16 °C by 2075. Annual precipitation would increase by 2% by 2030 and 10% by 2075 from 1990. | Roy, Rahman [56] |
| A decline in the number of cold days and cold nights. Increase in the number of heavy rainfall days during the monsoon season. | Badsha, Kafi [57] |
| Increase in heavy rainfall days and severe floods. | Sheikh, Manzoor [58] |

For every 1 °C temperature rise, irrigated and rain-fed rice yield is reduced by 11.1% and 14.4%, respectively, in Indonesia. For a 2 °C temperature rise, low land and high land rice production is reduced by 40% and 20%, respectively [59]. Statistical models and observational experiments have shown that a temperature increase of 1% could lead to an average 3.2% decrease in global rice yields. By the end of the twenty-first century, sustained temperature increases are expected to reduce global rice yields by 3.4–10.9% [60].

### 5.2. Sea Level Rise and Salinisation

Sea level rises are a consequence of rising global temperatures and are potentially the most serious impact of climate change in coastal Bangladesh [61]. The potential impacts of sea-level rise include increased salinisation, progressive inundation, coastal erosion, higher and more frequent storm surges, flooding, loss of wetlands, storm surges, and subsequent rice yield and production losses [62,63]. According to a World Bank Report [64], sea-level rise over the Bangladesh coast is predicted to be about 0.1 m by 2020, 0.25 m by 2050, and 1 m by 2100 compared to the base year (2000), resulting in the inundation of 17.5% of total agricultural (9.13 M ha) and non-agricultural land (3.82 M ha) by 2100. Aman rice, as well as subsequent dry-season crop cultivation in low-lying coastal areas, will be seriously affected by this inundation [36,65]. Analysis of tidal water fluxes over 30 years shows rising trends of water level in the Ganges tidal floodplain of 7 to 8 mm/year; the trend is 6 to 10 mm/year in the Meghna Estuarine floodplain and 11 to 21 mm/year in the Chittagong coastal plain areas [66]. Soil and water salinisation are also pressing problems in coastal regions. SRDI [67] estimated a 0.74% average increase in soil salinity per year, and, during the last 36 years, salt-affected areas have increased by 26.7%. Almost 6 million coastal people are already exposed to soil and water salinity and, by 2050, progressive salinity could affect the lives of 13.6 million people [36].

Coastal rice farming, in particular, was predicted to be seriously affected by sea-level rise and subsequent saline water intrusion and inundation [61]. Water management is a critical and complex issue for coastal and delta agriculture, and it is assumed that this issue will be further exacerbated by sea-level rise [68]. Rice is the main crop in coastal districts and supplies 21% of the national domestic rice production [18]. Crop diversification is currently limited in coastal districts due to a lack of fresh irrigation water, waterlogging, poor drainage facilities, salinisation, and natural disasters, as well as a range of socio-economic factors such as gender, access to credit, and household adaptive capacity [69]. The adoption rate of irrigation-dependent Boro rice and other dry-season crops in the coastal region is slower than in other parts of the country due to salinisation in groundwater and surface water [70].

### 5.3. Major Climatic Extreme Events and Stressed Rice-Growing Ecosystems

The rice-growing areas in Bangladesh are vulnerable to different types of climatic stresses and extreme events. Table 6 summarises the vulnerable rice-growing regions and associated climatic stresses. Major stressed ecosystems for rice cultivation are drought-prone north-west regions, low-lying haor basins, and saline south-west coastal zones. These three stressed ecosystems constitute 50% of the total rice area and contribute to more than 50% rice of the national rice harvest [18]. Although monsoon flooding is not new, the timing and level have increased in recent years [52]. Furthermore, pre-monsoon flash flooding has been a concern for Boro rice harvesting in Haor areas [30]. Groundwater drawdown poses a problem in the north-western districts of Bangladesh [71]. Other climatic stresses occurring in vulnerable rice-growing areas include heat stress in upland areas and coastal regions, extreme weather events, and earth subsidence [72].

Hydrology and water resource use has been affected by climate change, causing a scarcity of freshwater and posing a threat to Bangladesh's agriculture and livelihoods. According to Sarker, Alam [73], 1.5 to 2.0 M ha of irrigated rice will suffer some degree of water scarcity by the year 2025. Importantly, this will affect the Boro season, which contributes 55% of total rice output [18].

**Table 6.** Major climatic stresses and extreme events in vulnerable rice-growing regions in Bangladesh. Source: Author's calculation using data from the Department of Agricultural Extension [74], Soil Resource Development Institute (SRDI), and Bangladesh Agricultural Development Corporation [75].

| Major Climatic Stress | Vulnerable Rice-Growing Regions | Proportion of Total Rice Area Affected (%) |
|---|---|---|
| Monsoon flood | Coastal districts, Haor basins, Brahmaputra basin | 45 |
| Drought | North-western districts and coastal zone | 45 |
| Salinity | Coastal districts | 20 |
| Heat stress | North-western and central upland districts | 20 |
| Cyclone, storm | Coastal districts | 20 |
| Subsidence | Coastal districts | 15 |
| Groundwater drawdown | North-western districts | 32 |
| Early flash flood | Haor basins | 11 |

*5.4. Impacts of Environmental Change on Rice Production and Farming Systems*

The potential impacts of climate change on crop production and farming systems have been reported by many researchers. Relevant articles on the impacts of climate change on crop production are summarised in Table 7. In different climate scenarios, yield loss of Aus, Aman, and Boro rice, as well as wheat and potato, has been accounted for using different models. Rice yield loss due to environmental change is an important consideration for future rice security in Bangladesh.

**Table 7.** Impacts of climate change on rice production in Bangladesh.

| Impacts on Production | Models Applied | Source |
|---|---|---|
| HYV rice yield will be 15.6 per cent lower in coastal Upazila, where soil salinity will be greater than 4 dS/m by 2050 | Econometric yield model | Dasgupta, Hossain [76] |
| Yield loss of 10.5% for Aus rice, 10.8% for Aman rice, and 20.8% for Boro rice during 2040–2069. | CERES, MIKE21 | Ruane, Major [77] |
| A 33% reduction of average rice yields for 2046–2065 and 2081–2100 in Rangpur, Faridpur, and Barisal. | ORYZA2000 | Karim, Ishikawa [78] |
| Yield loss of wet season rice will be 0.2–5% by 2050 and 2–10% by 2070. Yield loss of early wet season rice will be 2–26% by 2050 and 5–26% by 2070. The yield increase of DS rice will be 3–10% by 2070. | DSSAT | Hussain [79] |
| The average yield gap of the BR3 variety was forecasted at 30, 43, and 52% for 2030, 2050, and 2070, respectively, and the corresponding yield gap of the BR14 variety was predicted at 37, 49, and 58%. | DSSAT | Basak, Ali [80] |

Climatic factors have an important effect on farming systems, crop choices, cropping patterns, and biotic stresses from natural enemies [81]. The increasing temperature, unexpected rainfall patterns, and climatic extreme events can potentially impact soil preparation, seedling raising, timely sowing/transplanting, water management, weed management, insect and disease management, harvesting, and postharvest processing. These impacts prompt farmers to shift to new farming systems or modify the existing ones for adaptation. Over the last couple of decades, rice farming systems have been showing a gradual shift from rice to non-rice crops, although rice remains dominant.

The population dynamics and community interactions of natural enemies of rice and other crops, including insect pests, plant pathogens, and invasive plant species, are predicted to shift in response to climate change. Growth, survival, multiplication, transmission, and the outbreak of diseases (fungi, bacteria, and viruses), pests, and weeds are impacted by changes in temperature, humidity, and precipitation [81].

## 6. Socio-Economic Challenges for Resilient Rice Production Systems

The resilience of the rice system is not only affected by climatic factors but is also influenced by several socio-economic factors, such as population pressure, farming experience, the land tenure system, farmer category, access to credit and extension services, peer-group pressure, household consumption, labour availability and income status, income potential, and consumer behaviour [82].

Currently, sufficient food is available to feed the global population. However, about 12% of the world's population is chronically undernourished in terms of energy intake. On the other hand, more than 1.4 billion adults above the age of 20 years are overweight. The projected increase in the world population from 7.4 billion in 2017 to 9.7 billion in 2050 is a big challenge for global food security [83].

Shrinking arable land is a challenge for the resilient rice system in Bangladesh [50] and many other developing countries. In Bangladesh, the average rate of annual agricultural land loss is 0.68% of total arable land due to urbanisation and non-agricultural use [84]. Globally, agricultural land use change due to population pressure, industrialisation, urbanisation, and unplanned expansion of aquaculture and pasture is considered a potential challenge for a sustainable farming system and food security [85,86].

Farm labour shortages due to increased non-farm employment opportunities and urban migration are a challenge for sustainable intensification of labour-intensive rice farming [12]. Farm mechanisation can address this issue, although mechanised transplanting, weeding, and harvesting are still limited due to the high price and limited availability of machinery, and rice farming, therefore, remains labour-intensive [87].

Demand for non-rice crops, medicinal plants, ornamental plants, and livestock is increasing with the growing consumer demand, which, in turn, places increasing pressure on rice-growing areas to be converted to other food systems [46]. Shrimp and prawn aquaculture in coastal Bangladesh and commercial orchard development in rice fields in the northern and south-western districts have significantly reduced the rice area [88]. The area under rain-fed Aus rice is continuously reducing due to low profitability, availability of more profitable crops (e.g., jute, maize, vegetables, spice crops, tuber crops), and reduced rainfall in the early wet season [89].

Geopolitical issues such as transboundary water sharing can also impact the intensification of cropping systems. For example, dry-season river water flow is reduced due to the construction of the Farakka Barrage, located upstream of the India-Bangladesh border, limiting dry-season rice cultivation in south-west coastal Bangladesh [90]. Trade restrictions, export bans, tariffs, and non-tariff barriers might impact the resilience of the rice system, destabilising rice availability and the market [91]. Disease pandemics, regional conflicts, and political crises also impact food security [92].

## 7. Market and Price Stability for Resilient Rice-Based Food System

A resilient food system is a priority for developed and developing countries, to achieve the United Nations Sustainable Development Goals (SDG 1, SDG 2). However, stability in food security is still a global concern, despite steady production growth over the last twenty years. Continuous population pressure, pandemics (e.g., COVID-19, SARS, Ebola), climatic disasters (e.g., droughts, floods, cyclones), political crises, and widespread pest attacks (e.g., locust infestation, late blight disease) are major drivers for destabilising the market of foods [93]. As rice is a widely consumed food crop, its price in the domestic and international markets has big food security implications. The stability of the rice market is dependent on various supply and demand variables, which are influenced by regional, national, and global factors [94]. At a local level, contracts can increase output price, and profitability for farmers and cooperatives may be beneficial for managing input costs, but it is unclear which marketing models are the most effective [95].

In recent times, rice farming has been a low-income or even loss-incurring business, discouraging farmers from intensification of rice systems. The wide difference in farmgate price and retail price deprives rice farmers of expected farm income. Over the last ten

years (2010–2020), rice price in the retail market has increased by 72–94%; however, the farmgate paddy price trend is almost flat despite the increasing cost of production [96]. Despite fertiliser subsidies for rice farmers, the cost of production is rising due to the increasing price of rice seeds, micronutrients, pesticides, fuel, and farm labour [97]. The production cost per kilogram of rice has been reported to have increased by 30% over the last ten years [98,99]. Farm labour wage has increased by more than double over the last ten years [87].

In Bangladesh, despite a good surplus of rice, access for low-income people is constrained. Food accounts for about 70% of total expenditures for the poor, and the poor spend at least 35% of their income on rice [100]. Therefore, a large spike in rice prices is a serious threat to this vulnerable group [101]. The Bangladesh government applies some policy tools and tariff mechanisms to keep the rice price affordable at the consumer level and also fair for farmers [5]. At the end of the financial year (FY) 2018/19, the government nearly doubled (from 28% to 55%) import tariffs to boost producer prices after rice production hit an all-time high of 36 Mt in FY 2017/18 [5].

## 8. Adaptation Strategies and Policy Support

Adaptation is an ecological, social, or economic response to actual or expected climatic stimuli and their effects or impacts [102]. Adaptation is an ongoing process in farming, and it is a critical strategy for reducing the severity of environmental and socio-economic impacts on agricultural value chains and farmer livelihoods. Table 8 summarises the common adaptation practices for sustaining rice production. Farmers have adopted a range of agronomic adjustments to their farming practices, including the use of stress-tolerant varieties, adjusting sowing dates, modifying irrigation schedules, and increasing the adoption of conservation agriculture practices, strategic nutrient management, integrated pest management, and multiple cropping. Some of the agronomic practices are now bundled and branded together as adaptation packages, such as 'climate-smart agriculture', although site-specificity and incorporation of local practices are important for adoption by farmers [103].

Autonomous adaptations are spontaneous, self-initiated actions undertaken to reduce risks in farming and livelihoods [103], while planned adaptations are technologies, programmes, or specific actions disseminated through systematic extension activities by non-farmer actors such as government agencies, NGOs, and the private sector [104]. In the literature, autonomous adaptations were more frequently assessed or reported than planned adaptations for farmers in coastal and northern Bangladesh. Nearly 75% of autonomous adaptation practices were agronomic- and farming systems-related, and there was also some adaptation through livelihood diversification, e.g., local off-farm work and migration [105]. The planned adaptations reported were focused on agronomic and farming systems responses, as well as infrastructure development (e.g., renovation of canals and sluice gates and construction of embankments). In many cases, farmers adopted both autonomous and planned adaptation practices (Table 9).

**Table 8.** Summary of adaptation strategies commonly reported in the literature for rice-based farming systems in Bangladesh.

| Type | Adaptation Measures |
|---|---|
| | *Biophysical adaptations* |
| Agronomic adaptation | Use of salinity-, drought-, submergence-tolerant crop varieties |
| | Use of short-duration high-yielding crop varieties |
| | Efficient water management |
| | Adjustment of sowing time |
| | Fertiliser and pest managements |
| | Relay and mixed cropping |

**Table 8.** *Cont.*

| Type | Adaptation Measures |
|---|---|
| Farming system adaptations | Practising a new cropping system, e.g., rotations<br>Crop diversification<br>Homestead gardening and rearing livestock |
| Infrastructure and mechanisation | Construction of embankment, dam, and water reservoir<br>Repairing sluice gates<br>Re-excavation of rivers, canals, and tributaries<br>The mechanisation of rice farming<br>Reallocation of household labour<br>Off-farm employment<br>Business opportunities |
| *Socio-economic adaptations* | |
| | Reallocation of household labour<br>Off-farm employment<br>Business opportunities |
| *Policy and contextual adaptations* | |
| | Economic tools, e.g., subsidies and rebates<br>Research, development, and extension, e.g., training, experimentation<br>Land zoning and legislative protection against industrial use of rice field<br>Public procurement policy and minimum price support |

**Table 9.** Agronomic and farming systems adaptation strategies to environmental and socio-economic challenges to rice production in Bangladesh. Note: AA = Autonomous Adaptation, PA = Planned Adaptation NW = north-west, SW = south-west.

| Autonomous and Planned Adaptation Options | AA | PA | Locations | Source |
|---|---|---|---|---|
| Shifting the planting date of Boro rice | x | x | NW Bangladesh | Acharjee, van Halsema [106] |
| Novel dry-season non-rice crops<br>Changed fertiliser use and sowing date | x<br>x | x<br>x | Khulna | Kabir, Cramb [105] |
| Reduce T. Aman area<br>Increased Aus and non-rice crop area<br>Delay in Aus transplanting time and early non-rice crops | x<br>x<br>x | <br>x<br> | NW Bangladesh | Kabir, Cramb [107] |
| Sorjan farming [1], relay cropping, and rainwater harvest<br>Salinity and flood-tolerant crop varieties, use of pheromone trap | <br>x | x<br>x | Patuakhali | Rashid [108] |
| Varietal improvement of rice | | x | Satkhira | Radanielson, Gaydon [109] |
| Cultivation of non-rice crops and rice-shrimp farming<br>Salinity-tolerant rice cultivation | x | x | SW Bangladesh | Kabir, Cramb [110] |
| Rice-prawn-vegetable farming system | x | | SW coastal Bangladesh | Ahmed and Diana [111] |
| Cultivation of non-rice crops<br>Homestead gardening<br>Wage work and non-farm activities | x<br>x<br>x | <br>x<br> | Khulna | Saroar [112] |
| Crop diversification and intercropping<br>Undertaking off-farm and non-farm wage work<br>Changing crop varieties | x<br>x<br>x | <br>x<br> | Satkhira | Mondal, Paul [113] |

[1] The *sorjan* cropping system is a series of constructed raised beds and lowered sinks, where dryland crops can be grown on the beds and wetland ones can be produced simultaneously in the sinks.

Farming system adaptations with modification, intensification, and diversification have been reported as a sustainable adaptation strategy. Farmers have started the cultivation of alternative annual crops (e.g., vegetables, legumes), forage crops, and perennial species (e.g., mango, guava) to offset the impacts of climate change [114]. Mixed farming systems and polyculture have also become more common with aquaculture (mainly shrimp

and prawns) and some livestock husbandry [115]. Infrastructure adaptations include embankment, barrages, dams, roads, re-excavation, repair of drainage canals, dredging of rivers, and setting up buried irrigation and drainage pipes [116].

Livelihood adaptations commonly include the enterprise diversification strategies noted above (e.g., livestock and aquaculture options) and the adoption of wage work and non-farm activities. These adaptation strategies are driven by multiple socio-economic and bio-physical factors, of which climate change pressures are assumed to be significant, though not necessarily the primary contributing factor.

Institutional barriers can potentially retard the effectiveness of adaptation initiatives undertaken by the government body, NGOs, and development partners [117]. Therefore, institutional capacity development is necessary to enhance the expected outcomes, whether for autonomous or planned adaptations.

As a signatory of the United Nations Framework Convention on Climate Change, Kyoto Protocol, Paris Agreement, and other climate-related treaties, Bangladesh has responded to climate change challenges relatively quickly [118]. Key outcomes have been to formulate the National Adaptation Programme of Action in 2009 and the Bangladesh Climate Change Strategy and Action Plan (BCCSAP) in 2009, and, subsequently, the Bangladesh Climate Change Trust Fund (BCCTF) was created from its resources in 2010. The food security issue has been placed first among the six pillars of BCCSAP. The adaptation to climate change has gained due importance in policy documents such as the National Agricultural Policy 2018 [119], Bangladesh Delta Plan 2100 [36], and Seventh Five-Year Plan (2015–2020). The government has emphasised research and extension policies for sustainable rice systems in Bangladesh. Incentives for promoting rain-fed Aus rice, subsidies in fertilisers and farm machinery, and rebates for electricity bills for irrigation are notable policy initiatives to sustain the rice system in Bangladesh [120].

The adaptations found in Bangladesh are often similar to those in other rice-producing countries in Asia. The use of new or well-adapted rice cultivars is common, including to address wet-season submergence and increasing dry-season salinity [121]. However, the adoption rate in Bangladesh can be lower than in India, Cambodia, and Nepal [122]. Whole-farm adaptations to address agronomic and labour pressures have included direct seeding as a popular adaptation option in Cambodia, Nepal, Thailand, and Sri Lanka—with labour savings of up to 80% compared with transplanting [122]—rice intensification, which is being practised in Vietnam and Pakistan for higher productivity, adaptation, and mitigation [123]. However, adoption of these techniques in Bangladesh has been slower due to several factors, such as the diverse socio-economic backgrounds of farmers, risk aversion for new farming technology among small and medium farmers, and inadequate extension services [44,115,124,125]. Other countries have been promoting private-sector partnerships, sponsored research and collaboration for digital technology solutions, smart extension services, and healthier rice programmes for climate-smart rice farming systems [126], while Bangladesh's progress in these areas is relatively limited [124]. To harmonise progress, International Rice Research Institute has recommended regional collaboration and cooperation to strengthen rice partnerships, exchange of genetic resources, value-chain, and sustainable resource management [127].

## 9. Research, Development, and Extension

The review recorded a range of impacts and adaptations concerning the environmental and socio-economic challenges to a resilient rice-based food system in Bangladesh. This section discusses the research, development, and extension needs identified from the examined literature, as summarised in Table 10.

**Table 10.** Summary of research and development needs for resilient rice systems in Bangladesh.

| Domain | Research, Development, and Extension Needs |
|---|---|
| Biophysical factors | • Climatic forecasting capacity for rainfall patterns and extreme events<br>• Breeding and selection of stress-tolerant varieties (e.g., salinity, drought, submergence)<br>• Developing effective farming options for dry-season crops<br>• Optimising mixed farming systems (e.g., agro-aquaculture)<br>• Irrigation and drainage technologies and improving water-use efficiency<br>• Mechanisation, from field preparation to post-harvest processing<br>• Infrastructure such as embankments, irrigation facilities |
| Socio-economic factors | • Current, localised understanding of the impact of demographic changes (e.g., gender, ageing population, migration)<br>• Market-based solutions at the local and regional level<br>• Rice trading and marketing in Bangladesh and globally<br>• Profitability and risks analysis of alternative adaption options |
| Policy and contextual factors | • Evaluation of institutional capacity at multiple levels, e.g., village-based Agricultural Officers, Upazila extension staff, scientific staff in research institutions<br>• (Re-)designing sustainable rice policies<br>• Developing (and evaluating) realistic participatory adoption pathways, in the context of supporting autonomous adaptation by farmers<br>• Understanding of socio-cultural issues, including women's participation in farming<br>• Maintaining an understanding of emerging challenges to rice security<br>• Enhancing farm mechanisation<br>• Ensuring farmers' access to the fair price |

Given the ongoing changes in the environment and socio-economic challenges, a thorough and up-to-date understanding of impacts on agricultural intensification options, agronomic and economic sustainability, and rural household livelihoods is needed, including issues regarding information flows and decision-making for households and the private sector. Region-specific research and extension planning and policy support are needed to sustain the rice system in the coming years [46].

In the context of declining freshwater resources, farming systems research for irrigated dry-season crops, particularly in coastal and north-western Bangladesh, should give special attention to increasing cropping intensity and livelihood opportunities. With different cropping combinations, agro-aquaculture needs to be optimised for sustainable coastal agricultural planning, addressing economic and environmental criteria [111]. Recent hydrological challenges have made irrigation (surface and groundwater) management issues more complicated, affecting irrigated Boro rice. A mechanised rice system is increasingly important to address labour scarcity. Therefore, research and development in surface and groundwater management and mechanisation need special attention.

Several types of autonomous and planned adaptations are practiced in the region; however, low adoption rates suggest improving extension and engagement strategies with farmers [17]. Over the last 20 years, several stress-tolerant rice varieties and modern technologies have been promoted to farmers for climate change adaptation [128]. The level of adoption and the performance and limitations of these stress-tolerant rice varieties should be assessed and investigated for further research, better-targeted extension, and subsequent policy recommendation [17]. To develop and disseminate climate-resilient adaptation technology, public and private institutions play a vital role. For a resilient rice system, the institutional capacity building of NGOs and government organisations is a vital requirement. However, due to a lack of research on policy impacts, evaluating institutions' performance and capacity is a challenge [129].

Several socio-economic and contextual issues pose further challenges to rice production in Bangladesh. Demographic changes, such as population increases, out-migration, and urbanisation, have impacted labour availability and cost, as well as rural household incomes [17]. Value-chain factors such as market stability and changing consumer demand have posed challenges for farming households to adapt their farming systems to minimise risk and maximise income and for equitable participation by women and poorer farmers [17,130]. Socio-economic constraints commonly vary between locations and situations [131]. Therefore, a locally specific understanding of the drivers of adaptation should underpin effective adaptation plans [132].

In addition to the impacts of climate change, rice farming households have gradually been leaving their traditional homelands, creating a significant problem for agriculture by reducing the availability of labour for cultivation, harvesting, and processing [133]. Demographic change and agribusiness change needs to be assessed and analysed to determine the most appropriate technological and market-based solutions at the local and regional levels and design sustainable rice system policies at higher levels of governance and management.

The retail price of rice is an important strategic issue for socio-economic development and stability in Bangladesh. The domestic and international market for rice is strongly affected by extreme climatic events. Production losses in major rice-producing regions due to climatic changes and events have the potential to destabilise the global market and apply uneven pressure on rice consumers [43,134]. Research on rice-based farming systems with present status, profitability, risks, sustainability, and impending challenges can create new pathways for sustainable rice production in Bangladesh [48].

An up-to-date understanding of socio-cultural issues is also essential, especially for factors that influence the adoption and successful implementation of climate change adaptation by farmers and locals [135]. In particular, gender equality remains a challenge globally, and women's participation in decision-making and action is no less constrained in Bangladesh [136]. Strong information systems and policies will be required to ensure the involvement and coordination of stakeholders for adaptation.

## 10. Summary and Policy Recommendations

A resilient rice-based food system is high on Bangladesh's national policy agenda for food self-sufficiency and international commitments to sustainable development goals. However, ensuring a reasonable price for consumers and producers is crucial to achieving these broad goals. This study reviewed the literature relating to critical environmental and socio-economic changes that are significantly impacting rice systems in Bangladesh. Based on the initial conceptual framework presented in Figure 1, the impacts, adaptation, and policy responses have been summarised and discussed, and the preparedness among diverse stakeholders, including rice farmers, government organisations, and NGOs in Bangladesh, has been highlighted. In Figure 3, the conceptual framework has been expanded to include specific key challenges and adaptation measures affecting the resilient rice-based food systems in Bangladesh that were identified in this review.

It was noted that increased rice production will be needed to meet rising domestic consumer demand. Intensification of rice production (and food production in general), with efficient soil, fertiliser, water, and pest management, is imperative to maintaining sustainable growth in national rice production, given reducing arable land for several reasons. Hydrological and water management issues must be addressed in a sustainable manner. Research, management, and extension activities should be given priority for the most vulnerable rice-growing regions of Bangladesh (coastal zone, north-west region, and Haor areas) to reduce the spatial yield variation. Anomalous rainfall patterns are likely to impact rain-fed Aus and Aman rice. Likewise, irrigated Boro rice is affected by declining river flows and the drawdown of groundwater. Given the environmental vulnerability, government and private sector stakeholders have an opportunity to provide research and extension, as well as policy support, to sustain rice production. Location-specific

rice-based farming systems should be planned and implemented, with adaptation programmes mainstreamed into existing institutions and planning processes carried out locally and nationally.

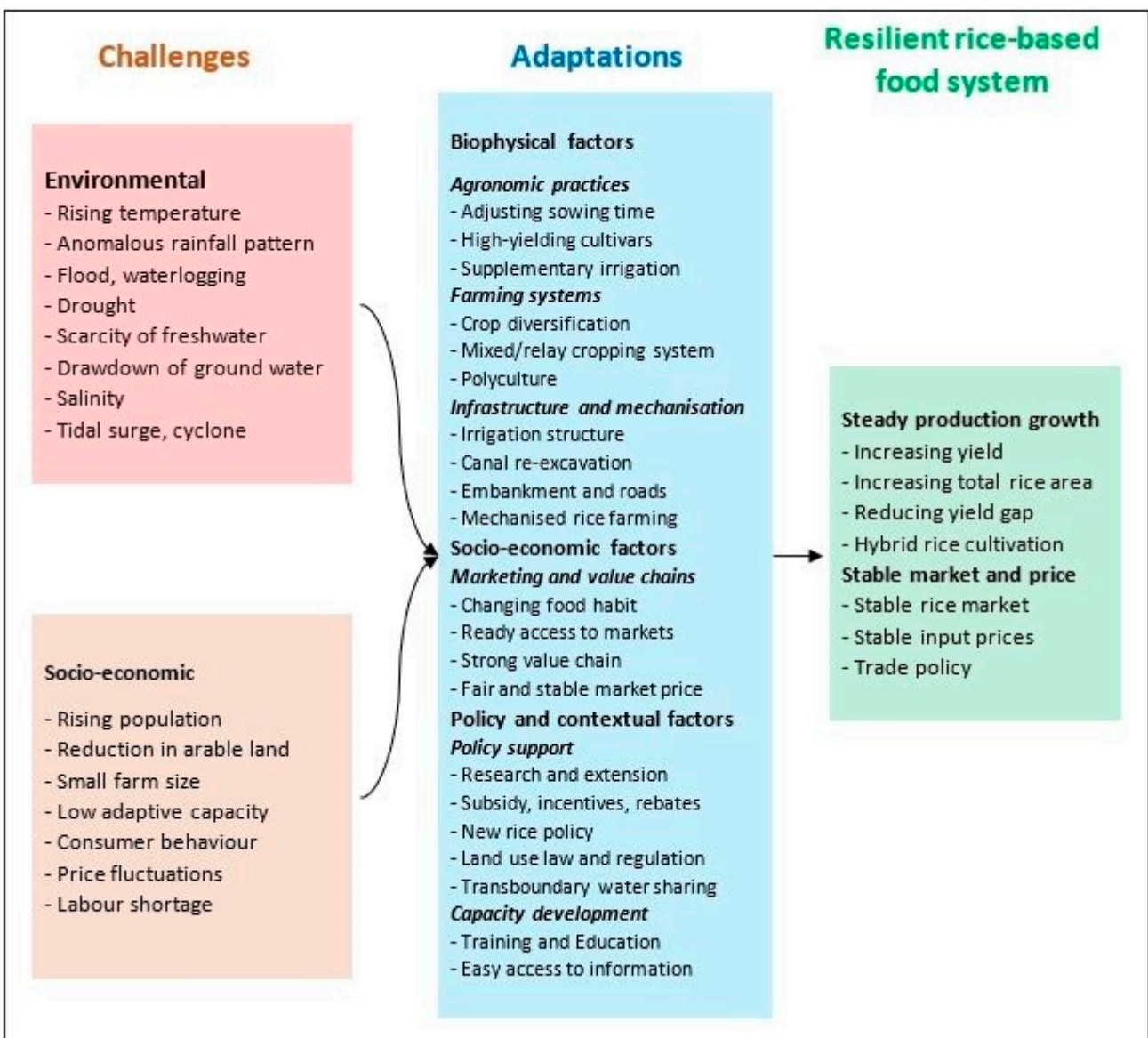

**Figure 3.** Conceptual framework of key challenges and adaptation measures affecting resilient rice-based food system in Bangladesh.

Policy initiatives have largely been diverted to infrastructure development and comprehensive disaster management. However, a focus on policy that addresses effective locally relevant farming systems research, knowledge management, institutional capacity building, and supporting farmer-driven adaptation (e.g., appropriate extension) is recommended. Labour shortages and increasing cost pressures highlight the need for greater support for farm mechanisation, although mechanisation poses a new set of challenges. The ongoing development of the adaptive capacity of farmers, extension workers, input dealers, and rice traders is necessary and should be locally relevant and acknowledge existing local experience and adaptation initiatives. Given the government's broad focus on infrastructure and other planned activities, farmers' autonomous adaptation initiatives are significant. Thus, there is a role for regional and national government agencies, as well

as NGOs and private sector actors, to promote and support farmer-initiated adaptation activities and enhance the adoption of planned adaptations to ensure resilient rice systems.

**Author Contributions:** Conceptualisation: M.R.J., P.K., M.J.K. and L.L.d.B.; Writing—original draft preparation: M.R.J. and P.K.; Writing—review and editing M.R.J., P.K., M.J.K. and L.L.d.B. All authors have read and agreed to the published version of the manuscript.

**Funding:** The first author acknowledges the scholarships provided by the University of New England, Armidale, Australia.

**Data Availability Statement:** Sources for secondary data are cited in the manuscript.

**Acknowledgments:** The first author is highly grateful to the Ministry of Agriculture, Government of the People's Republic of Bangladesh, for granting deputation for the PhD programme.

**Conflicts of Interest:** The authors declare that they have no conflict of interest.

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
