# Peer review of "Challenges and Adaptations for Resilient Rice Production under Changing Environments in Bangladesh"

_land, doi:10.3390/land12061217_

Round 1

Reviewer 1 Report

Rice is one of the most widely grown crops in the world, with dynamics in climate and soil conditions widely reported to be causing disruptions in its  production systems. To meet up with ever increasing demand for it, rice producers have been taking measures to adapt to the changing situations. These adaptation practices however tend to vary with geographic contexts in which the production is taking place. Little has been done by the authors to compare the situations in Bangladesh with those of areas outside the Asia region. Readers would benefit well from this article if they are provided with information on the extent to which the situations in Bangladesh compare or remarkably differ from those of other areas. There is the therefore the need for the authors to use the experiences of other rice producing areas outside the Asia region to evaluate the extent of dynamics in climatic and edaphic effects on rice production in Bangladesh can be utilized in promoting rice production in other regions of the world.

The quality of use of English language in the manuscript is generally okay even though it has been produced by non-native English speakers. There are nonetheless some minor changes that need to be carried out on grammar and expressions.

Author Response

Minor editing of the English Language required

RESPONSE: Done: Whole MS

Compare Bangladesh’s adaptation to other rice-producing countries (in Asia and outside Asia)

RESPONSE: Done. New paragraph added comparing adaptation in other countries (mostly South and SE Asia): L431-444

Reviewer 2 Report

Comments to the Author

1. Please use official method of writing the title of your manuscript write first capital letter of each word

2. Add take home message at the end of the abstract
3. Abstract portion has a lot of grammar and spelling mistakes

4. Data provided in form of survey should be represented graphically  

5. Whole manuscript needs to be justified

6. References portion contains much of the mistakes kindly provide data with appropriate cited literature

Overall Language

English language needs a through revision from some professional service or native speaker because it contains a lot spelling and grammar mistakes

I will suggest the author to work on above mentioned shortcomings otherwise it can be rejected if not revised properly 

Author Response

Please use the official method of writing the title of your manuscript write the first capital letter of each word

RESPONSE: Done. Title case used: L1-2

Add a take-home message at the end of the abstract

RESPONSE: Done. Take-home message added to Abstract: L24-25

The abstract portion has a lot of grammar and spelling mistakes

RESPONSE: Done. Corrected by a native speaker

Data provided in the form of the survey should be represented graphically

RESPONSE: We are unclear what this is referring to? Please clarify

The whole manuscript needs to be justified

RESPONSE: Done. All pages have justified page alignment

The references portion contains much of mistakes. Kindly provide data with appropriately cited literature

RESPONSE: Done. Reference list was proof-read and revised

The English language needs a thorough revision from some professional service or native speaker because it contains a lot of spelling and grammar mistakes

RESPONSE: Done. Language has been revised by native speakers

Reviewer 3 Report

This manuscript is suitable for publication if they are able to address the following main concern in the abstract, methods, results and conclusion section. 

Manuscript Title: Challenges and adaptations for resilient rice production under changing environment in Bangladesh

Land] Manuscript ID: land-2416910

General comments: This manuscript is suitable for publication if they are able to address the following main concern in the abstract, methods, results and conclusion section.

Abstract: The authors should provide the methodology in the abstract section: What methodology adopted for this review paper? Is it systematic review or meta-analysis or integrative reviews? If the authors used meta-analysis, the use of statistical methods should be introduced in short and precise.

How many articles has been downloaded? How to select 154 articles for final review? I think the number of articles or documents are too high (154). I recommend the authors to minimize the number of citations. Use the most important one and ignore the others. Article inclusion and exclusions techniques can be introduced under methodology section.

Introduction:

It is sound if the authors give high attention to climate change. Climate change has high impact on crop production in general and rice cultivation in particularly. I understand that the authors introduce the issue of extreme weather events in abstract section and the problem of global warming and SDG-13 in introduction section, but this is not sufficient.

Although, the authors review about rising temperature, rainfall irregularity weather extremes and other climate change related problems, they do not report any quantitative figures as far the impact of climate change on rice is concerned. They are not able to link the impact of climate change on rice production i.e., the information provided is too general ….for better understanding the authors can provide some quantitate data. For instance, what will happen if the average temperature of a particular region or area is increased by 1°C on rice production. Just this is an example.

Methodology: Line 142 it is better if the authors included some of the keywords used for this search engine from Web of Science and Scopus.

Figure 4 is not in the right place.

Conclusion: Should be to the point.

References: 150 references are too high (I recommended the author to focus on the most important literatures).

Thus, based on my assessment, the paper can be considered for publication, if they are able to addressed all comments and concern above.

Author Response

The English language is fine. No issues were detected.

RESPONSE: OK

The authors should provide the methodology in the abstract section

RESPONSE: Done. Methodological information provided in the Abstract: L15-17

How many articles have been downloaded? How to select 154 articles for final review? I think the number of articles or documents are too high (154). I recommend the authors to minimize the number of citations. Use the most important one and ignore the others. Article inclusion and exclusion techniques can be introduced under the methodology section.

RESPONSE: Done. The methodology section has been expanded to explain this: L132-136, 144-150

I recommend the authors to minimize the number of citations.

RESPONSE: Done. The number of references has been reduced to 135

Give high attention to climate change in the introduction

RESPONSE: Done. Climate change impacts have been further highlighted in the Introduction: L45-48

Put quantitative data on the impacts of climate change on crop yield

RESPONSE: Done. Figures added to the text: L244-248

Line 142 it is better if the authors included some of the keywords used for this search engine from Web of Science and Scopus

RESPONSE: Done. Search terms added to the Methods: L132-136

Figure 4 is not in the right place

RESPONSE: There is no Figure 4?

Conclusion: Should be to the point

RESPONSE: The title of that section has been changed from “Conclusion and policy recommendations” to “Summary and policy recommendations” in order to emphasise its purpose as a ‘discussion’ section.

References: 150 references are too high

RESPONSE: Done. The number of references has been reduced to 136

Round 2

Reviewer 2 Report

All the comments/suggestions has been incorporated the authors. 

A native English-speaking or professional English editing service must edit before publication of this manuscript.

Reviewer 3 Report

I have no more comments.